# Inflammation Responses to Bone Scaffolds under Mechanical Stimuli in Bone Regeneration

**DOI:** 10.3390/jfb14030169

**Published:** 2023-03-21

**Authors:** Junjie Wang, Bo Yuan, Ruixue Yin, Hongbo Zhang

**Affiliations:** 1School of Mechanical and Power Engineering, East China University of Science and Technology, Shanghai 200237, China; 2Spine Center, Department of Orthopaedics, Shanghai Changzheng Hospital, Second Affiliated Hospital of Naval Medical University, Shanghai 200003, China

**Keywords:** inflammation, mechanical environment, bone regeneration, ultrasound, mechanical force

## Abstract

Physical stimuli play an important role in one tissue engineering. Mechanical stimuli, such as ultrasound with cyclic loading, are widely used to promote bone osteogenesis; however, the inflammatory response under physical stimuli has not been well studied. In this paper, the signaling pathways related to inflammatory responses in bone tissue engineering are evaluated, and the application of physical stimulation to promote osteogenesis and its related mechanisms are reviewed in detail; in particular, how physical stimulation alleviates inflammatory responses during transplantation when employing a bone scaffolding strategy is discussed. It is concluded that physical stimulation (e.g., ultrasound and cyclic stress) helps to promote osteogenesis while reducing the inflammatory response. In addition, apart from 2D cell culture, more consideration should be given to the mechanical stimuli applied to 3D scaffolds and the effects of different force moduli while evaluating inflammatory responses. This will facilitate the application of physiotherapy in bone tissue engineering.

## 1. Introduction

Bone is an important tissue in the human body. It has the ability to self–heal, but external means of promoting repair are still needed when it suffers from large–scale defects [1]. Current treatments for bone defects include autologous bone grafting, allogeneic bone grafting, and bone engineering scaffolds [2,3,4]. However, the first two are prone to secondary injury, poor size matching, and immune rejection during transplantation, so their clinical application is limited [5]. To address these issues, bone scaffolds manufactured using various techniques, such as 3D printing technology, have been widely used [6]. In addition, 3D bioprinting technology can be used to produce bone implants that are personalized according to the needs of patients, promoting osteogenesis while regulating immune responses [7]. The body inevitably has an immune response to implants, as appropriate immune responses can protect the body from foreign pathogens; however, severe immune reactions, such as immune rejection, may lead to implantation failure. Therefore, the current focus of bone tissue engineering is on how to reduce immune rejection while retaining the ability to promote osteogenesis, which places certain demands on the properties of bone scaffolds.

The ideal bone implant has good biocompatibility, biodegradability, biostability, and mechanical properties [5]. Biocompatibility means that the implant should not be cytotoxic and should not inhibit the adhesion, proliferation, and differentiation of bone cells. Degradability refers to the fact that the scaffold should degrade at the same rate as the mineralized tissue deposition. If the degradation rate is too fast, it will not provide support before the new bone matures; conversely, it may exacerbate the foreign–body–rejection response. Biological stability refers to the implant’s ability to resist biological aging. An ideal bone implant should maintain sufficient structural stability during prolonged immersion in body fluids and blood and should not undergo crosslinking or phase changes. The bone scaffold should also have sufficient mechanical properties to provide support for bone reconstruction until the new bone matures. To meet these requirements, several biomaterials have been developed for 3D printing. Materials used for 3D bioprinting mainly include natural and synthetic polymers [8]. Natural polymers include chitosan, sodium alginate (SA), collagen (COL), gelatin (GEL), hyaluronic acid (HA), and cellulose [9]. Their most important feature is excellent biocompatibility, in addition to which they have some unique advantages [5]. For example, chitosan has been shown to have good osteoinductive ability. SA is able to increase the viscosity of polymer solutions. COL is best suited to mimicking the extracellular matrix (ECM). GEL has strong cell–adhesion properties. HA supports cellular structures and acts as a lubricant. Cellulose is very widely found in nature. However, these natural polymers have the disadvantage of lacking sufficient mechanical strength [10]. Synthetic polymers include polylactic acid (PLA), polycaprolactone (PCL), polycarbonate (PC), polyetheretherketone (PEEK), and polypropylene (PP). PLA has excellent mechanical properties but poor tensile strength. It tends to produce an acidic environment during metabolism, leading to inflammation. PCL has a long degradation cycle and can provide long–term protection to the tissue. PC has high tensile strength and no cell toxicity, but it tends to absorb moisture from the air, causing surface defects. Unlike PC, PEEK has very poor water absorption, allowing it to stay in a watery environment for long periods of time. PEEK is also highly resistant to high temperatures and can be autoclaved without affecting its material properties, but this same property limits its use in 3D printing because the printing temperature must be turned up to a high level. PP has the lowest density among synthetic polymers and a longer service life, but it has a slightly lower yield strength compared with PCL. In summary, natural polymers have better biocompatibility and degradability, while synthetic polymers have excellent mechanical properties [11].

A major concern with bone scaffolds prepared by 3D printing technology is that the performance of the scaffold is greatly limited by the bioink that is used. For example, gelatin is very sensitive to temperature and dissolves at high temperatures; PP shrinks during the printing process due to crystallization, and this is not conducive to printing [12,13]. Therefore, an effective strategy for bone tissue engineering is to prepare composite scaffolds of natural–synthetic materials to exploit the advantages of both. Another strategy is material modification. Materials used for modification can be inorganic, metal ions, or plasma. Inorganics such as hydroxyapatite can significantly enhance the thermal stability of PEEK. The bending modulus of hydroxyapatite–modified PEEK is increased by nearly 30% compared with pure PEEK [14]. Metal ions are commonly used to modify metal alloy implants, such as Mg^2+^, Zn^2+^, Cu^2+^, Co^2+^, and Li^2+^. Galli et al. [15] placed threaded implants coated with mesoporous TiO_2_ films loaded with Mg^2+^ in the tibia of rabbits and found that the local release of magnesium ions significantly promoted the implant binding to the rabbit tibia. Liu et al. [16] found that degradable magnesium–copper alloy enhances the viability of MC3T3–E1 cells and promotes bone formation. Yusa et al. [17] found that zinc–modified titanium implants promote the osteogenic differentiation of human mesenchymal stem cells (hMSCs). Plasma modification is commonly used to improve the surface properties of biomaterials, such as wettability, roughness, and surface free energy. Kostov et al. [18] modified polyethylene (PE) and polypropylene (PP) with Ar gas by means of cold atmospheric pressure plasma jets to obtain polymers with higher roughness and wettability. Jorda–Vilaplana et al. [19] improved the surface free energy and roughness of PLA by atmospheric plasma modification. Kozelskaya et al. [20] improved the biocompatibility of titanium implants with the use of micro–arc oxidation

A big challenge in regenerative medicine is immune rejection. Immune rejection caused by bone scaffolds mainly includes chronic inflammation, foreign body giant cell formation, and fibrous capsule formation [21]. Here, we focus on the inflammatory response induced by the bone scaffold, including the release of inflammatory factors and the polarization of macrophages. Macrophages are myeloid immune cells that are distributed in almost all kinds of tissues. Their main role is to regulate the inflammatory response and to degrade and phagocytose dead cells or foreign bodies. During the early stages of implant placement, macrophage recruitment and differentiation can induce a favorable inflammatory response and reduce the risk of wound infection. After acute inflammation has passed, however, the formation of foreign body giant cells (FBGCs) induces a chronic inflammatory response. In the late stage of implantation, macrophages are also able to promote tissue repair and angiogenesis. Therefore, the latest scaffolding strategies are focused on how to modulate macrophage behavior to regulate the inflammatory response.

The inflammatory response, characterized by redness, swelling, warmth, and pain, is the body’s defensive response to stimulation by toxins and pathogens [22]. A proper inflammatory response can destroy invading germs and protect the body from damage. However, excessive inflammatory responses may induce inflammatory diseases such as osteoarthritis, muscle wasting, and neurotrauma [23,24]. These inflammatory diseases place enormous distress and financial burden on those affected. In the process of inflammatory response, signaling pathways play an important role. When cells are stimulated by external stimuli, proteins on the cell membrane sense the stimulation and transmit signals into the cell. Membrane proteins receive signals and initiate a series of enzymatic reactions. This process is known as signaling–pathway conduction. Common signaling pathways of inflammation include the PI3K/AKT, JAK/STAT, TLR, NF–kB, MAPK, and NLRP3 inflammasome pathways. In the process of signaling, cytokines are involved in the transmission of information as messengers. Cytokines include interleukin (IL), tumor necrosis factor (TNF), and interferon (IFN), among others [25]. Based on their role in inflammation, cytokines can be classified into pro–inflammatory and anti–inflammatory types. Pro–inflammatory factors include IL–1, IL–6, IL–12, IL18, TNF–α, and IFN–γ, while anti–inflammatory factors include IL–10, IL–13, etc. [26]. IL–1β and TNF–α cytokines are currently considered to be those with the greatest impact on inflammation. IL–1β is thought to be associated with activation of the TLR, NF–kB, and MAPK pathways. It affects downstream pathways by binding to the membrane receptor IL–1R1, followed by binding to MyD88. TNF–α can bind to two different membrane receptors: TNF–R1 and THF–R2. The TNF–α––TNF–R1 complex can then activate the NF–kB pathway. Similar to the response to IL–1β, the MAPK pathway is activated during this process. In this review, we focus on the TLR, NF–kB, MAPK, and NLRP3 inflammasome pathways and analyze how mechanical stimulation affects the release of inflammatory factors via these pathways. The relationship between these four pathways is shown in Figure 1.

Bone tissue engineering refers to the combination of natural or synthetic scaffolds, osteoblasts, and cytokines into a whole, and bone scaffolds should have mechanical properties and biocompatibility similar to those of natural tissue in order to promote bone tissue regeneration [27,28]. This strategy has been used in a variety of bone–related diseases, such as osteoporosis, fractures, and osteoarthritis. However, in the actual repair process, fibrous connective tissue often occupies the bone defect, hindering normal bone formation. This fibrous tissue has a much lower mechanical strength than normal bone, resulting in the formation of defective bone. Current studies have shown that appropriate force stimulation can simulate the mechanical environment of human bone growth and promote the proliferation and differentiation of osteoblasts. Such physical stimulation (cyclic strain, ultrasound, electromagnetic, fluid shear force) has already been shown to play positive roles in bone regeneration in vitro and in vivo [29,30,31]. Notably, the mechanical environment not only affects the proliferation and differentiation of osteocytes but also plays a regulatory role in the immune microenvironment. This article mainly discusses the effects of different mechanical stimuli on osteogenesis, in connection with scaffolding strategies to explore inflammation and its signaling–pathway mechanisms operating under mechanical stimuli.

## 2. Signaling Pathway

### 2.1. TLRs Signaling Pathway

Toll–like receptors (TLRs) are transmembrane proteins that are widely expressed in mammals. They represent the body’s first line of defense, recognizing external signals to regulate cell proliferation, differentiation, apoptosis, and inflammatory responses. Current research shows that the TLR family contains a total of 13 TLR proteins, 11 of which are present in the human body [32]. Of these, TLR4 can recognize lipopolysaccharide (LPS), the main component of Gram–negative bacterial biofilm, leading to activation and downstream signal transmission. Most signaling relies on MyD88. TLR recruits MyD88 through the TIR domain, promoting IRAK4 (IL1RI–related protein kinase) binding and phosphorylating IRAK1 [33]. The interacts with TRAF6 to form the IRAK1––TRAF6 complex. This complex can further activate the NF–kB pathway, thereby releasing inflammatory factors.

### 2.2. NF–kB Signaling Pathway

NF–kB includes two main subunits, p50 and p65, which, together, form the NF–kB/Rel complex. Inhibitory–kB (IkB) is mainly found free in the cytoplasm and binds to NF–kB to inhibit its activity [34]. As shown in Figure 2a, the TLR signaling pathway can transmit external stimuli to the NF–kB pathway in a cascading manner and activate IkB kinase (IKK) to promote IkB phosphorylation. Activated NF–kB translocates to the nuclear coding region to regulate the expression of pro–inflammatory mediators, such as COX–2, iNOS and cytokines [35]. The NF–kB signaling pathway is an extremely important pathway in the inflammatory response and plays an active role in the expression of numerous inflammatory factors. At the same time, the NF–kB pathway can also regulate other signaling pathways, such as the MAPK and NLRP3 inflammasome pathways.

### 2.3. MAPK Signaling Pathway

Mitogen–activated protein kinase (MAPK) is a family of serine/threonine protein kinases with a three–level signaling cascade: MAPK kinase kinase (MEKK), MAPK kinase (MEK), and MAPK [36]. Figure 2b shows the activation process of the MAPK pathway. MAPK mainly includes three kinases: (1) extracellular signal–regulated kinase (ERK), (2) c–Jun N–terminal kinase (JNK), and (3) p38 MAPK. Ras is a GTP–binding protein, and pro–inflammatory stimuli convert GDP to GTP to activate Ras and, thereby, phosphorylates activating MEKK, such as Raf. MEKK activates and phosphorylates MEK, and phosphorylated MEK then activates and phosphorylates MAPK, further regulating cell proliferation, differentiation, apoptosis, and inflammatory responses. ERK1/2, JNK, and p38MAPK are confirmed to be closely associated with the inflammatory response [37].

### 2.4. NLRP3 Inflammasome Signaling Pathway

The term “inflammasome” was first coined by Tschopp to describe a high–molecular–weight protein complex formed in the cytosol [38]. It is mainly used to recruit certain caspases, such as casp1, casp4, and casp5. These caspases act as inflammatory caspases and are mainly involved in the inflammatory response of cells. Among them, casp1, as the main platform for pro–IL–1β and pro–IL–18 proteolysis and cleavage, promotes the maturation and secretion of IL–1β and IL–18 [39,40].

The NLRP3 inflammasome is a special kind of inflammasome, mainly comprising the NLRP3, ASC adaptor, and pro–casp1 proteins [41]. Their structures are shown in Figure 3a. NLRP3 protein consists of a pyrin domain (PYD), a nucleoside–binding domain (NBD), and leucine–rich repeats (LRRs). LRRs are thought to maintain the autoinhibited state of NLRP3 protein, while NBD is responsible for binding to adenosine triphosphate (ATP) [42]. When stimulated by danger signals, the NLRP3 inflammasome can be activated to promote the secretion of pro–inflammatory cytokines [43].

The activation of the NLRP3 inflammasome is divided into two steps: (1) the priming stage, in which endogenous or exogenous signals lead to the synthesis of NLRP3 and pro–IL–1β; and (2) the activation stage, in which NLRP3, ASC, and pro–casp1 assemble into the NLRP3 inflammasome, which, in turn, promotes IL–1β and IL–18 secretion [44]. The protein expression level of NLRP3 is critical for inflammasome assembly, and the high expression of NLRP3 is thought to increase the activation rate [45,46]. When stimulated by exogenous or endogenous signals, NLRP3 binds to the ASC adaptor protein through the PYD domain, and the CARD structure on ASC then recruits pro–casp1 to form the NLRP3 inflammasome [25], as shown in Figure 3b. Pro–casp1 undergoes self–cleavages into activated casp1 and promotes the maturation and secretion of IL–1β and IL–18 [47]. It has been shown that NLRP3 inflammasome activation is not only affected by the NF–kB pathway but also by oxidative stress responses, such as those induced by reactive oxygen species (ROS) [38,48].

## 3. Physical Stimulation in Bone Regeneration

Bones are constantly subjected to mechanical loading during movement. According to Wolff’s law, the structure and strength of bone is affected by the applied mechanical force [49]. A common obstacle in bone tissue engineering is that fibrous connective tissue rapidly occupies the bony defect, precluding the occurrence of normal bone formation (osteogenesis) [50]. This connective tissue has low mechanical strength, which can lead to bone defects. This challenge has been addressed via the physical stimulation of bone cells (e.g., cyclic stress), which is considered to promote bone cell growth as much as possible and to inhibit connective tissue formation [51].

The process of physical stimulation for bone formation can be divided into two phases: mechanotransduction, where physical signals are converted into intracellular biochemical signals; and signaling, where biochemical signals regulate the proliferation and differentiation of osteoblasts through signaling pathways.

### 3.1. Mechanosensors in Mechanotransduction

The role of mechanosensors is to recognize and transmit mechanical signals. Common mechanosensors include integrins, ion channels, and primary cilia [52].

Integrins are transmembrane proteins that transmit forces to the cytoskeleton and from the cell to the extracellular matrix. The integrin complex consists of α and β subunits, of which β1 has good anti–inflammation and osteogenesis effects. The mechanotransduction effect of integrins is mainly exerted through focal adhesions (Fas). Fas are macromolecular protein complexes formed by integrins and intracellular linker proteins, such as focal adhesion kinase (FAK). Once formed, Fas can establish connections between the extracellular matrix and the cytoskeleton, thereby transmitting mechanical signals.

Ion channels play key roles in osteogenic mechanotransduction. These can align with other connexin–displaying cells, forming a functional connection known as a “gap junction” [52]. Ca^2+^ can enter and exit between osteocytes and osteoblasts through gap junctions (calcium channels), which transmit mechanical signals [53].

Piezo1 is the protein with the highest number of transmembrane structural domains and is highly conserved across species [54]. Upon sensing a mechanical force, the central pore of Piezo1 opens and allows for the passage of cationic ions (e.g., Ca^2+^, Na^+^, K^+^, Mg^2+^, etc.). Piezo1 regulates YAP–dependent type II and type IX collagen expression in osteoblasts, thereby regulating osteoclast differentiation and bone resorption. Li et al. treated mature mice with Piezo1 agonists and found significant upregulation expression of bone formation markers (COX–2, Wnt1, etc.) [55].

### 3.2. Ultrasound

Ultrasound is a special sound wave with a frequency greater than 20 kHz that can cause local oscillations between particles, thereby transmitting mechanical force. Therefore, ultrasound is recognized as one of the physical stimuli for bone regeneration engineering. As a kind of ultrasound, the effect of low–intensity pulsed ultrasound on cartilage repair has been confirmed by a large number of studies [56,57]. Low–intensity pulsed ultrasound (LIPUS) is an ultrasound with a frequency of 1–3 MHz and a SATA intensity of 0.02–1 W/cm^2^ [58]. As a form of physical therapy, it is characterized by being non–invasive, causing little damage to the tissue, and the thermal effect is also very limited due to its low intensity. The therapeutic effect of LIPUS is attributed to its non–thermal effects, mainly cavitation, acoustic flow, and acoustic radiation forces [59]. Due to its unique superior qualities, ultrasound therapy has increasingly become an important means of bone healing in recent years [60].

In in vitro experiments, researchers have found that ultrasound can enhance the expression of the osteogenic marker OCN [61]. To further elucidate the mechanism by which ultrasound promotes osteogenesis, numerous studies have focused on the ultrasound activation of mechanosensing–integrin proteins [62]. Activated integrins bind to focal adhesions, which serve as a link between the cytoskeleton and the extracellular matrix for these activated integrins [63]. Studies have shown that ultrasound can activate focal adhesion kinase (FAK), which, in turn, activates the PI3K/AKT pathway [64,65], as shown in Figure 4a. In addition, as the most important signaling pathway regulating cell proliferation, differentiation, and apoptosis, the MAPK pathway has been shown to be activated by ultrasound [66]. Carina et al. [67] found that ultrasound can upregulate the expression of MAPK1 and MAPK6, confirming the effect of ultrasound on the MAPK pathway. Under the combined action of the two signaling pathways, ultrasound significantly upregulates the expressions of osteogenic factors such as RUNX2, OCN, and OSX [52]. However, the signaling–pathway mechanism of ultrasound–promoted osteogenesis is still poorly understood and needs to be further studied.

### 3.3. Cyclic Stress

Cyclic stress mainly refers to cyclic compression and cyclic stretching. Cyclic stress induces compression and relaxation of the ECM, leading to strain in the cartilage, mimicking the cyclic strain that the human body experiences on bones during daily activities [50]. The magnitude of the strain is also critical for cartilage regeneration. The magnitude of tensile strain is reportedly associated with the inhibition of adipogenesis, whereas the introduction of rest in the strain has no apparent effect on osteocyte growth [68]. New research shows that cyclic strain can upregulate the expression of paladin (an actin–associated protein) through the Wnt signaling pathway, promoting stem cells to differentiate into bone and cartilage rather than fat [69]. Together, these indicate that cyclic strain plays an important role in maintaining tissue homeostasis and promoting cartilage regeneration.

The ECM–integrin pathway is currently being widely discussed regarding the mechanism by which cyclic strain promotes osteogenesis [50]. When cells are mechanically stimulated, force signals need to be converted into biochemical signals within the cell, and integrins, as transmembrane receptors, provide such a platform. Integrins realize the transformation of force signal into a biochemical signal by connecting the extracellular matrix and intracellular skeleton elements (actin filaments, non–muscle myosin, and associated proteins), thereby maintaining the dynamic balance between tension and compression [70,71]. As shown in the Figure 4b, integrin–mediated focal adhesion kinases (FAKs) and Src tyrosine kinases can activate the MAPK and NF–kB pathways, thereby promoting the secretion of osteogenic markers Runx2, BMP–2/4, OCN, and Osx [72]. On the other hand, mechanical stimulation can activate calcium channels on the cell membrane and induce an extracellular calcium influx to increase calcium concentration, thereby promoting bone healing [73,74]. Mechanical stimuli that affect Ca^2+^ signaling in chondrocytes include compression, fluid flow, hydrostatic pressure, and osmotic pressure [75,76]. Due to the piezoelectric effect of bone, cyclic stress can also create a potential difference between the inside and outside of the bone and affect bone healing. Normally, when a bone is stretched, the concave side (compressed) is negatively charged, and the convex side (stretched) is positively charged, which causes the bone to grow more on the compressed side and degrade more on the stretched side [77].

### 3.4. Other Physical Stimuli

Normal activities in the human body will generate a certain electric field, usually in the range of 10–60 mV [78]. This physiological electric field (EF) plays an important role in regulating the physiological balance of cells and tissues. When the tissue is damaged, EF can guide migrate of cells to the wound and promote wound healing; when EF is inhibited, however, the rate of wound healing is significantly dampened [79]. Studies have shown that a certain degree of electrical stimulation promotes bone regeneration [80,81]. In in vivo and in vitro experiments, electrical potential has also been shown to play an important role in the proliferation, differentiation, and migration of osteocytes [82,83]. Therefore, EF is increasingly used for bone healing.

Similar to EF, another method of physical stimulation involves using a pulsed electromagnetic field (PEMF), which is generated by an unstable electrical current in a coil [84]. A certain frequency of electromagnetic stimulation has been shown to regulate the cell cycle of MSCs, promote their proliferation and differentiation, and improve the osteogenic stimulation of ASCs [85]. Under PEMF stimulation, increased TNF–β1 and BMP–2/4 expression was detected in osteoblasts [86,87], and the intracellular calcium ion concentration increases [88]. This suggests that PEMF may exert osteogenic effects through a similar mechanism to that of mechanical stimulation.

### 3.5. In Vivo and In Vitro Experiments

Osx is a transforming factor involved in the differentiation of preosteoblasts to mature osteoblasts that has an important role in the pre–osteogenic phase. Suzuki et al. [89] treated rat osteoblasts with LIPUS (1.5 MHz, 20 min, 30 mW/cm^2^) for 15 days and found a significant increase in Osx expression. Another transcription factor that plays an important role in the process of osteogenesis is Runx2. Similarly, upregulation of Runx2 expression in response to LIPUS has been reported [90]. Previous studies have shown that the effect of LIPUS on osteoblast proliferation is controversial. Hasegawa et al. [91] reported that LIPUS significantly enhanced osteogenic differentiation but had no significant effect on osteoblast proliferation. Gleizal et al. [92] showed increased proliferation of primary cranial cap osteoblasts in mice after LIPUS treatment. 

In vivo experimental studies based on animal models also support the osteogenic effect of physical stimulation. An experiment showed that LIPUS–treated male Wistar rats have higher levels of alkaline phosphatase (ALP) expression [93]. This suggests that LIPUS can be used as an adjunct to consolidate bone repair. Wang et al. [94] reported that 20 min/d of cyclic mechanical stimulation promoted the proliferation of rabbit anterior maxillary chondrocytes, demonstrating the sensitivity of chondrocytes to mechanical signals in vivo. Meulen et al. [95] studied the response of cancellous bone to mechanical stimulation by using a rabbit loading model. It was shown that mechanical stimulation can inhibit osteoporosis by increasing the bone volume fraction.

## 4. Inflammation under the Mechanical Environment in Bone Scaffolds

### 4.1. The Immune Environment of Bone Scaffolds

The immune rejection caused by bone scaffold transplantation mainly consists of inflammation, foreign–body giant cell (FBGC) formation, and fibrous envelope formation. As shown in Figure 5, when the bone scaffold is implanted in the human body, protein molecules are first recruited near the scaffold, followed by the formation of a blood–based transient provisional matrix [96]. The provisional matrix is rich in growth factors and cytokines that recruit immune cells to the injured surface and prompt the release of inflammatory cytokines [97]. Then an acute inflammatory response is induced, marked by the recruitment of neutrophils. After acute inflammation, monocytes and lymphocytes signal the onset of chronic inflammation [98]. During chronic inflammation, macrophages may come together to form mononuclear giant cells called FBGCs. FBGCs can degrade biological materials on scaffold surfaces, thereby affecting biocompatibility. After the inflammatory response subsides, tissue–repair cells can be recruited to the damaged surface and promote tissue regeneration. This process may result in the formation of granulation tissue, which is later transformed into a fibrous envelope. Fibrosis is an obstacle to bone regeneration engineering. A large amount of fibrous tissue rapidly occupies the bone defect and hinders bone regeneration. 

In the process of immunity, the phenotype of macrophages plays a critical role in the success or failure of bone implants. Macrophages were previously considered the first line of defense against microbial infection, but their roles in wound healing are also being elucidated. Macrophages are derived from monocytes, as shown in Figure 6. Monocytes leave the bone marrow and enter the bloodstream under the action of colony–stimulating factors (CSFs) secreted by stromal cells in the blood, where they are recruited to the tissue surface by chemical inducers and differentiate into macrophages [99]. Macrophages acquire the ability to fuse through the activated DAP12/Syk pathway to form FBGCs [100]. However, this process is highly dependent on the platform of monocyte adhesion and the type of surface blood protein. Macrophages can be divided into “classically activated” M1 macrophages and “alternatively activated” M2 macrophages. M1 macrophages can be activated by a variety of cytokines, including IFN–γ, LPS, and TNF–α. In in vitro experiments, researchers have usually employed LPS+IFN–α to induce M1 polarization. Macrophage polarization also plays an important role in host defense by secreting pro–inflammatory cytokines such as IL–1β, IL–6, and IL–12. Recent studies have further divided M2 macrophages into M2a, M2b, and M2c subtypes, where M2a is induced by IL–4 or IL–13, M2b is activated by immune complexes and agonists of TLR or IL1R, and M2c is activated by IL–10 and glucocorticoids. In general, M2 macrophages are considered to be anti–inflammatory. However, tissue fibrosis can also occur when the activity of M2 macrophages is misregulated [96]. Most macrophages in the early stage of inflammation are of the M1 type, and the transition from M1 type (pro–inflammatory type) to M2 type (anti–inflammatory type) indicates the resolution of the inflammatory response and contributes to the release of osteogenic factors and the formation of new bone [101]. Therefore, a suitable immune environment can promote the transformation of M1 macrophages to M2 type to suppress the inflammatory response during transplantation and then can finally achieve the purpose of repairing bone tissue.

### 4.2. Regulatory Strategies for the Immune Environment of Bone Scaffolder

There are many strategies targeting the inflammatory response induced by bone implants as a means to regulate the scaffold immune environment via inhibition of the inflammatory response. These mainly include surface modification, drug delivery, and mechanical stimulation [102].

#### 4.2.1. Surface Modification

The surface properties of biomaterials are critical for protein adsorption and regulate the immune microenvironment by mimicking the structure and properties of native tissues, determining the host immune response and the subsequent tissue–healing cascade [103]. The surface properties (stiffness, surface topography, wettability, etc.) also have an important influence on the polarization of macrophages [104]. The manner in which material stiffness affects macrophage polarization, however, remains controversial. Most studies confirm that the increase in stiffness is associated more with a pro–inflammatory phenotype. Sridharen et al. [105] used collagen–coated polyacrylamide gels to demonstrate that increased stiffness leads to pro–inflammatory polarization. In another study, however, the authors prepared polyacrylamide gel substrates with different stiffnesses. Contrary to Sridharen’s conclusions, highly stiff substrates were found to exhibit stronger M2 polarization in another study, where it was further elucidated that this effect is achieved through the NF–kB pathway [106]. It has been speculated that activation of the NF–kB pathway by some denatured adsorbed protein may possibly alter the cellular response to stiffness [107]. Although the mechanism of macrophage response to biomaterial stiffness is still unclear, stiffness control remains an important direction for surface modification of biomaterials.

Surface morphology also affects the macrophage phenotype. Zhang et al. [108] reported that macrophages are sensitive to changes in titanium surface roughness. M2 macrophage polarization was enhanced over a small range of roughness (Ra = 0.51–1.36 μm), while roughness outside of the range upregulated a mixture of pro– and anti–inflammatory markers. It has been shown that micropatterns play an important role in adjusting macrophage phenotype. Optimizing the size and density of micropatterns helps drive M1 macrophage polarization toward M2. Moreover, nanostructures have been reported to affect the phenotype of macrophages. In one study, nanostructured and submicron–structured titanium scaffolds reduced the expression levels of pro–inflammatory cytokines [109]. 

Modification to endow hydrophilicity by plasma methods and hydrolysis methods is a common method to enhance biomaterial accessibility to biological molecules [110]. The hydrophilicity of a material can affect the activity of macrophages. In macrophages cultured on hydrophilic materials, the expression of anti–inflammatory factor IL–10 is found to increase, while the expression of pro–inflammatory factors TNF–a, IL–1b, etc., decreases [111].

#### 4.2.2. Drug Delivery 

In recent years, the strategy of adding anti–inflammatory drugs to scaffold materials to obtain anti–inflammatory properties has been widely discussed. Ginsenoside Rb1, one of the main active components of ginseng, has been shown to have anti–inflammatory properties. Wu et al. [112] developed a silk––fibroin–gelatin porous scaffold (GSTR) loaded with Rb1 and TGF–b1 and observed that the IL–1β–induced inflammatory response was significantly inhibited. GSTR scaffolds have the ability to release anti–inflammatory drugs in a sustained manner, which can continuously deliver Rb1 and TGF–b1, creating an anti–inflammatory microenvironment conducive to cartilage regeneration. Liu et al. [113] designed a 3D bioprinted scaffold loaded with bone–marrow mesenchymal stem cells (BMSCs) by alternately printing kartogenin–loaded PCL and BMSC–loaded methacrylated hyaluronic acid (MEHA) to solve the problems of collapse and instability of traditional structures. Diclofenac sodium was loaded onto matrix metalloproteinase–sensitive polypeptide–modified MEHA as a major anti–inflammatory strategy, significantly reducing IL–1β expression.

#### 4.2.3. Mechanical Stimulation

Physical stimulation is currently finding widespread use in bone tissue engineering. Enhancing the osteogenic differentiation of MSCs by applying mechanical stretch (compression) stimuli to scaffolds is proven to be an effective therapeutic strategy. At the same time, physical stimulation also affects the physicochemical properties of the scaffold; for example, the loading of mechanical force significantly increases the degradation rate of the polylactic acid (PLA) scaffold [114]. On this basis, researchers have further explored the effect of physical stimulation on the inflammatory microenvironment of bone scaffolds. Recently, Piezo1 has been shown to be an important mechanosignaling protein involved in the regulation of immune cell activity under mechanical stimulation conditions and exerts different regulatory effects on inflammatory responses under different mechanical loading protocols [115]. Zhang et al. [116] designed a bioreactor and applied cyclic compressive stimuli to hydroxyapatite scaffolds placed within to determine the effect of mechanical stimulation on the inflammation of MSCs seeded on the scaffolds. Their results suggest that cyclic compression, as one of the physical stimulation methods, can effectively inhibit the secretion of inflammatory cytokines and regulate the anti–inflammatory microenvironment of the bone scaffold. This seems to indicate that physical stimulation can be used as a new anti–inflammatory strategy to counteract the scaffold–induced inflammatory response. However, their study did not further elucidate the mechanism of this effect.

### 4.3. Effects of Physical Stimulation on Inflammatory Response

In contrast to the effect of physical stimulation on the inflammatory response of cells on scaffolds, the anti–inflammatory effect of physical stimulation is not a new topic. Researchers globally have conducted extensive research on the effects of two physical stimuli, namely ultrasound and cyclic stress, on inflammation in order to determine their application prospects in the field of anti–inflammation.

#### 4.3.1. Ultrasound

The role of ultrasound in inhibiting the inflammatory response has been discussed [117,118,119,120], and researchers have further studied the effect of ultrasound on the inflammatory signaling pathway to determine the mechanism of inhibiting the inflammatory response. Zheng et al. [121] found that LIPUS (frequency, 1 MHz; duty cycle, 20%; pulse repetition frequency, 100 Hz; intensity, 0.5 W/cm^2^; and 20 min/d) induces caveolin–1 activation and inhibits the p38 and ERK phosphorylation, thereby inhibiting the expression of pro–inflammatory factors. Zhang et al. [122] reported that LIPUS could alleviate the expression of LPS–induced inflammatory factors, such as IL–1β, IL6, and IL8. In addition, they revealed that this effect was achieved by inhibiting the TLR4–MyD88 and NF–kB pathways. Sahu et al. [123] reported that continuous low–intensity ultrasound inhibits NF–kB activation induced by TNF–α. In the study by Chen et al. [124], LIPUS inhibited the expression of TLR4/NF–kB pathway–related proteins and, thereby, the secretion of downstream factors TNF–α, IL–1β, and IL–6. Nakao et al. [125] found that LIPUS (frequency, 1 MHz; pulse repetition frequency, 1 kHz; intensity, 30 mW/cm^2^) inhibits the formation of the TLR4/MyD88 complex and thereby activation of downstream p38, ERK1/2, and NF–kB pathways. However, their study did not elucidate how LIPUS inhibits the formation of the TLR4/MyD88 complex. Ueno et al. [126] found that LIPUS (frequency, 3 MHz; duty cycle, 20%; intensity, 0.5 W/cm^2^) could inhibit LPS–induced p38MAPK phosphorylation and muscle atrophy. Zhang et al. [127] found that LIPUS (1 MHz; pulse repetition frequency, 250 Hz) significantly inhibits the IL–1β–induced expression of NO, PEG2, and MMPs. In addition, LIPUS also inhibited the activation of the NF–kB pathway, thereby reducing the inflammatory response. Liao et al. [128] found that LIPUS (1.5 MHz; duty cycle, 20%; intensity, 30 mW/cm^2^; 20 min/d) could promote ECM synthesis, inhibit the inflammatory response, and inhibit NF–kB pathway activation. Xia et al. [129] found that the expression of NLRP3 protein on the BMSC cells of mice treated with ultrasound was significantly reduced, and the secretion of inflammatory factors such as IL–1β and IL–18 was inhibited. This suggests that ultrasound can inhibit the initiation phase of the NLRP3 inflammasome. The above results are shown in Table 1.

#### 4.3.2. Cyclic Stress

Cyclic stress has been shown to influence the maintenance of tissue homeostasis by modulating cellular functions, such as development, inflammation, bone remodeling, and tumor progression [130]. However, it has also been reported that sustained destructive mechanical stimulation may induce chronic inflammatory responses [131]. For example, excessive mechanical stimulation by mechanical ventilation can lead to lung inflammation [132]. Therefore, appropriate cyclic stress is required to maintain the homeostasis of the immune environment. Maruyama et al. [130] reported that cyclic stretching (20% strain, 10 cycles/min) significantly inhibits IL–1β secretion and further confirmed that it inhibits the NLRP3 inflammasome signaling pathway. However, their study showed that cyclic stretching has no effect on LPS–induced time–dependent degradation of IkB and NF–kB transcriptional activity. Therefore, they suggest that cyclic stretching inhibits the NLRP3 inflammasome pathway by affecting the adenosine monophosphate–activated protein kinase (AMPK) pathway rather than the NF–kB pathway. Iwaki et al. [133] found that cyclic stretching (20% strain, 50 cycles/min) promotes IL–8 expression in human pulmonary microvascular endothelial cells (HPMVECs) and enhances the inflammatory response. In addition, this process is achieved through the p38 MAPK pathway instead of ERK1/2 and JNK. The report by Oudin et al. [134] also supports this conclusion that cyclic stretching promotes IL–8 secretion through the p38 MAPK pathway. Sebag et al. [135] found that mechanical stretching (MS) inhibits LPS–induced keratinocyte–derived chemokine and tissue factor expression. The expression level of TLR4 was significantly lower in the LPS+MS group than in the LPS group, indicating that mechanical stretching inhibits the expression of TLR protein. Charles et al. [136] came to the opposite conclusion. Their study showed that the mRNA and protein expression levels of TLR2 were significantly increased in human lung epithelial cells (A549) when exposed to mild cyclic stretching (20% strain, 20 cycles/min for 24 h). The above studies demonstrate that cyclic stretching has both pro– and anti–inflammatory effects depending on the parameters of cyclic loading and other external environments (e.g., LPS induction).

### 4.4. Physical Stimuli in Bone Tissue Engineering

The effect of physical stimulation on osteogenesis was discussed in the previous section. The anti–inflammatory strategies described in this section provide new ideas for the application of physical therapy in bone tissue engineering. Physical factors play a highly critical role in the life activity of cells. Compared with surface modification and drug delivery, physical stimulation may be a more direct and attractive approach. However, in applying physical stimulation, the form and parameters of physical stimulation must be carefully chosen. Different forms and parameters may lead to drastically different results. Furthermore, although the positive effects of physical stimulation in bone tissue engineering have been demonstrated in vitro, further in vivo experiments are needed for valid in vivo assessments. This is dependent on well–designed animal experiments and strict data validation.

## 5. Conclusions

This paper reviewed the effects of physical stimuli such as ultrasound and cyclic stress on osteogenesis and inflammation in bone regeneration engineering. It is concluded that mechanical stimulation improves the osteogenesis–promoting ability of bone implants and suppresses severe inflammatory responses, thus serving as guidance for the use of physical therapy in bone tissue engineering. However, the current physical therapies still have several shortcomings:

1. When considering cyclic stress as the force application method, most researchers considered cyclic tension. It should be noted that, while bone tissue is subject to a variety of mechanical forces, including pressure, strain, shear, and torsion, pressure is the predominant mechanical stimulus of these forces. It is, therefore, necessary to consider the effect of cyclic compression on the osteogenic and inflammatory effects produced by bone scaffold implantation in humans.

2. The most recent studies of mechanical stimulation on inflammatory response were mainly carried out on 2D cells rather than 3D scaffolds. Therefore, it is not clear how mechanical stimulation affects the immune environment and inflammatory response after scaffold implantation.

In this paper, the physical signals affecting the immune environment of bone scaffolds during transplantation were reviewed. The stimulation of ultrasound and cyclic stress on bone regeneration and bone immunity was discussed in detail, in particular, the manner in which the physical stimuli act on bone cells and the difference in the signal pathways between these two stimuli. This might help researchers build an appropriate mechanical environment of bone scaffolds to inhibit inflammatory response in order to improve the success rate of transplantation, thereby expanding the application of physical therapy in bone tissue engineering.

## Figures and Tables

**Figure 1 jfb-14-00169-f001:**
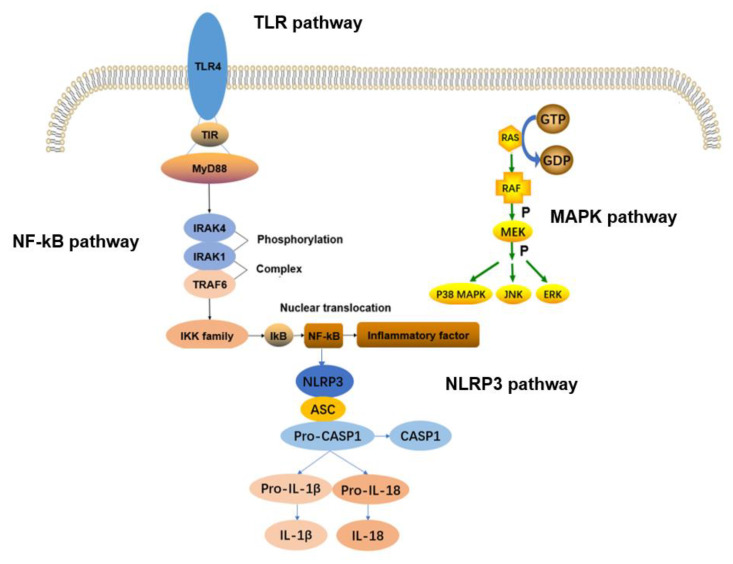
TLR pathway and MAPK pathway.

**Figure 2 jfb-14-00169-f002:**
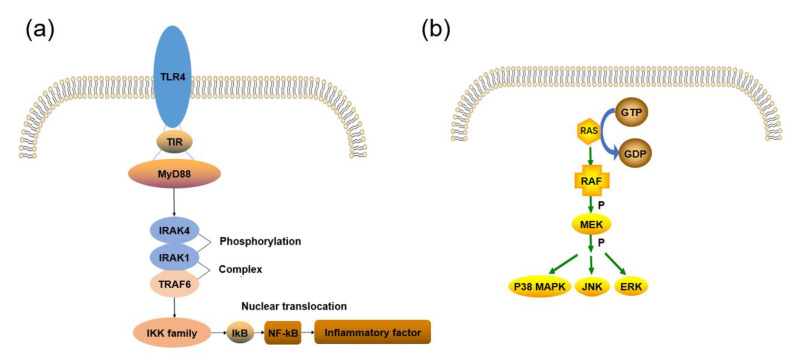
Schematic diagram of signaling pathways: (**a**) NF–kB and (**b**) MPAK.

**Figure 3 jfb-14-00169-f003:**
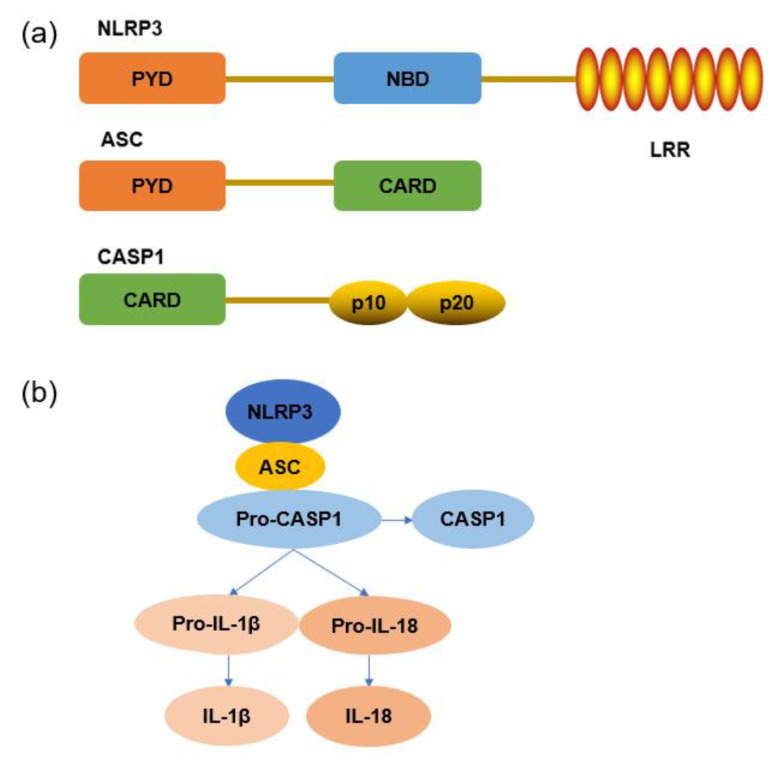
NLRP3 inflammasome: (**a**) protein structure and (**b**) signaling pathway.

**Figure 4 jfb-14-00169-f004:**
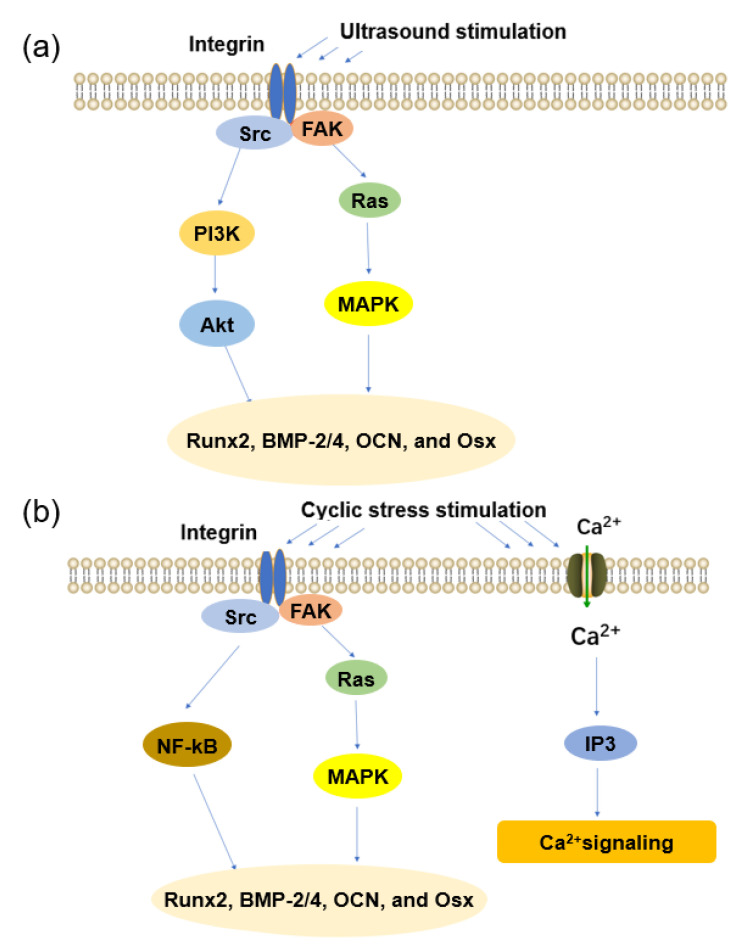
Mechanism of mechanical stimulation to promote osteogenesis: (**a**) ultrasound and (**b**) cyclic stress.

**Figure 5 jfb-14-00169-f005:**
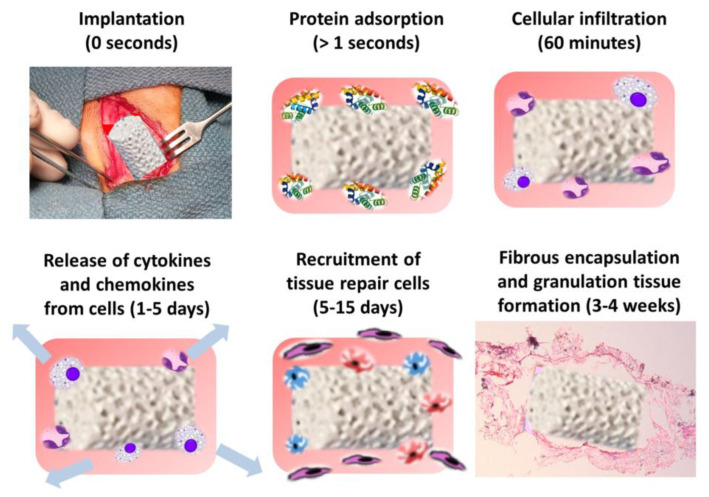
Implantation of the scaffold into the body; reprinted with permission from [96].

**Figure 6 jfb-14-00169-f006:**
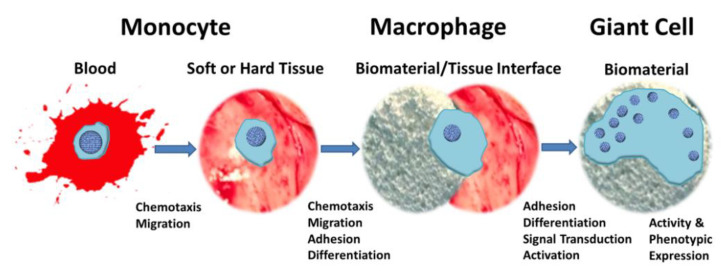
A schematic depiction of the transition of a monocyte to an FBGC; reprinted with permission from [96].

**Table 1 jfb-14-00169-t001:** Effects of ultrasound on inflammatory signaling pathways.

Author	Ultrasound Parameters	Conclusion
Zheng et al. [121]	Frequency, 1 MHz; duty cycle, 20%; pulse repetition frequency, 100 Hz; intensity, 0.5 W/cm^2^; 20 min/d	LIPUS induces caveolin–1 activation and inhibits the phosphorylation of p38 and ERK, thereby inhibiting proinflammatory–factor expression.
Zhang et al. [122]	Frequency, 1.5 MHz, pulse repetition frequency 1 kHz, duty cycle 20%; intensity, 10, 30, 60, and 90 mW/cm^2^	LIPUS reduces the expression of IL–1β, IL–6 and IL–8 by inhibiting TLR4–MyD88 and NF–kB pathways.
Sahu et al. [123]	Frequency, 5 MHz; continuous ultrasound; intensity, 0.528 W/cm^2^	cLIUS enhances cartilage phenotype and cell migration by inhibiting TNF–α–induced NF–kB pathway.
Chen et al. [124]	Frequency, 1 MHz; intensity, 0.528 W/cm^2^	LIPUS inhibits the expression of related proteins in the TLR4 and NF–kB pathways.
Nakao et al. [125]	Frequency, 1.5 MHz; pulse repetition frequency, 1 kHz; intensity, 30 mW/cm^2^	TLR4/MyD88 complex inhibits p38 and ERK1/2 phosphorylation in downstream pathways in addition to NF–kB pathway activation.
Mizuki et al. [126]	Frequency, 3 MHz; 20% duty cycle; 0.5 W/cm^2^ intensity	LIPUS inhibits LPS–induced p38 MAPK phosphorylation and muscle atrophy.
Zhang et al. [127]	Frequency, 1 MHz; pulse repetition frequency, 250 Hz; sound pressure, 0.1, 0.2, 0.3, and 0.4 MPa; time 1, 2, 3, 4, and 5 min/d	LIPUS inhibits NF–kB pathway activation.
Xia et al. [129]	Frequency, 8–18 MHz	The expression of NLRP3 protein was significantly inhibited, and the secretion of pro–inflammatory factors such as IL–1β and IL–18 was reduced.
Liao et al. [128]	Frequency, 1.5 MHz; duty cycle, 20%; intensity, 30 mW/cm^2^; 20 min/d	LIPUS promotes ECM synthesis, suppresses inflammatory responses, and inhibits NF–kB pathway activation.

## Data Availability

Not applicable.

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
