# Peer review of "Inflammation Responses to Bone Scaffolds under Mechanical Stimuli in Bone Regeneration"

_jfb, 2023, doi:10.3390/jfb14030169_

Round 1

Reviewer 1 Report

This review paper focuses on mechanical stimuli and their effect on inflammation response during bone generation. The contents cover several aspects of bone regeneration under mechanical stimuli, such as signaling pathway, types of stimuli, etc. The paper is well organized and has a clear layout. However, there are some minor mistakes that should be corrected.

Some grammatical mistakes and typos should be corrected, for example:

Page1 line 21, "self-heal" should be used instead of "self-healing" 

Page 6 line 165, delete "the"

page 12 line 409, typo: "Amoung" 

Page 13 line 413, "understanding" would be more appropriate than "learn" 

There are more such mistakes and the authors should proofread before resubmit.

Reviewer 2 Report

The review article entitled: Inflammation response of bone scaffolds under mechanical stimuli in bone regeneration present an interesting topic. Unfortunately, some major revision is necessary. The following issues should be addressed:

1.      1. The writing in this article must be checked, as some sentences are can be written shorten. For example, the first sentence in the abstract. It’s better to make 2 sentences out of it. A lot of this examples can be found.

2.      2. Two different citation styles were used. Moreover, space sings between words and the reference are missing.  

3.      3. In the introduction, the authors writing about 3D printing and mention not the disadvantages of 3D printed implants, as different surface modification approaches are necessary: For example, the surface modification of PEEK.  Also the surface modification of 3D printed metal alloy implants is missing: https://doi.org/10.3390/jfb13040285 Moreover, the differences between biodegradable and biostable polymer implants is not even mentioned. This point should be added, as it is a physical stimuli for the enhancement of bone regeneration.

4.     4. In the introduction, the importance of macrophages in the inflammation process with implants and the implant design is not mentioned and therefore missing.

5.      5. Space signs between value and unit are missing. Moreover, also space signs between values are missing particularly.

6.      6. The author’s contribution is missing. Please add.

7.      7. Are the figures self-drawn or taken from other sources? Are the figures drawn according to other figures? Please indicate this.

8.      8. Chapter 5 is nothing more than a part of the conclusion with an outlook and has nothing to do with a discussion. Put chapter 5 at the end of the conclusion.  

Reviewer 3 Report

human body. You can see a lot of work went into preparing this article. however, as reviewer I  found a few details that require clarification:

abstract

   This part of the work should also contain conclusions. Like many readers, at the beginning they read the abstract itself. if he finds it interesting, then he reads the whole article and can quote it. And we all want our work to be cited as often as possible.

Introduction

It seems to me that it would be good to cite a little more extensively which natural polymers are used as scaffoldings and which artificial ones in 3D technology. Give their positive and negative properties in more detail.

Su, C.; Chen, Y.; Tian, S.; Lu, C.; Lv, Q. Natural Materials for 3D Printing and Their Applications. Gels 2022, 8, 748. https://doi.org/ 10.3390/gels8110748

 Raszewski, Z.; Chojnacka, K.; Kulbacka, J.; Mikulewicz, M. Mechanical Properties and Biocompatibility of 3D Printing Acrylic Material with Bioactive Components. J. Funct. Biomater. 2023, 14, 13. https://doi.org/10.3390/ jfb14010013

I would suggest adding a table at the end of the article explaining all the abbreviations. It makes it much easier to write this interesting article.

Line 101

(Raf et al) at the ref is 25 line 113. But in Ref part 25 is

Zhang, Y.; Pizzute, T.; Pei, M. A review of crosstalk between MAPK and Wnt signals and its impact on cartilage regeneration. 485 Cell Tissue Res. 2014, 358, 633-649, doi:10.1007/s00441-014-2010-x.

Line 379   Kentaro et al[106] but in the ref part it is

Maruyama, K.; Nemoto, E.; Yamada, S. Mechanical regulation of macrophage function - cyclic tensile force inhibits NLRP3 682 inflammasome-dependent IL-1β secretion in murine macrophages. Inflamm. Regen. 2019, 39, 3, doi:10.1186/s41232-019- 683 0092-2

Mai et al[109] found that cyclic

Iwaki, M.; Ito, S.; Morioka, M.; Iwata, S.; Numaguchi, Y.; Ishii, M.; Kondo, M.; Kume, H.; Naruse, K.; Sokabe, M.; et al. Mechanical stretch enhances IL-8 production in pulmonary microvascular endothelial cells. Biochem. Biophys. Res. Commun. 2009, 389, 531-536, doi:10.1016/j.bbrc.2009.09.020.

References

Whitney, N.P.; Lamb, A.C.; Louw, T.M.; Subramanian, A. INTEGRIN-MEDIATED MECHANOTRANSDUCTION PATHWAY OF LOW-INTENSITY CONTINUOUS ULTRASOUND IN HUMAN CHONDROCYTES. Ultrasound Med. Biol. 2012, 38, 1734-1743, doi:10.1016/j.ultrasmedbio.2012.06.002.

-          please use lowercase letters when quoting the title of the article

good luck with further research

Reviewer 4 Report

The current manuscript reviews the inflammatory events elicited by bone scaffolds and mechanical stimuli under bone regeneration. While the topic of the review is relevant to the journal and presents great interest to the scientific community, to this reviewer this review lacks scientific rigor, accuracy and information for the Journal of Functional Biomaterials and the target audience. Therefore, the manuscript requires extensive improvement before being considered for publication.

Major suggestion are as follows:

11)      During the review, authors use the term “immune response” very vaguely. While the immune response to an implanted material is inevitable, it has been clear that materials that promote a favorable or immune response perform better to those that produce a immune acute reaction, chronic inflammation and/or encapsulation. Please revise all the manuscript and clarify these concepts.

22)      Page 2, first paragraphs. The information provided is outdated and limited. Line 55, there are many kinds of interleukins, please specify. Suggested references that may assist:

https://www.nature.com/articles/sigtrans201723   

33)      Since the review is based on mechanical stimulation, pathways that relate integrins, mechanosensors, PIEZO1, FAKs, YAP/TAZ, Hippo etc. are missing in the review. They are briefly mentioned afterwards, but given the topic of the manuscript they should be developed.

44)      Section 3. It is not clear if these strategies are designed to be followed in bioreactors, in vivo after implantation, etc. This should be clarified given the extensive work carried out in the field.

55)      Page 8. Last paragraph. The dichotomy M1/M2 is outdated and inaccurate. It is known that is very limited to describe the wide spectra of phenotypes that macrophages can adopt in response to biomaterials.

 https://link.springer.com/article/10.1007/s10439-021-02832-w

66)      M1 phenotype is achieved in vitro thanks to LPS and IFNg, but these are not the events that occur in vivo since LPS is a gram negative endotoxin, and these relates poorly to the mechanotransduction  topic. This should be clarified.

77)      Section 4.2.1. The information on effect of surfaces and physical properties on immune cell phenotype is very limited given the amount of work in the field. Besides there are very few examples specific to the bone regeneration area (which also has a great amount of work).

https://www.sciencedirect.com/science/article/pii/S1742706121002580?via%3Dihub

88)      Section 4.2.2. Seems more related to drug delivery strategies rather than composition adjustments or tailoring.

99)      Overall section 4 lacks of specific strategies aimed to bone.

Minor comments:

110)  Language, formatting and typos must be checked through the manuscript

111)  The quality of the figures could be improved

Round 2

Reviewer 2 Report

The authors of the review article "Inflammation responses to bone scaffolds under mechanical stimuli in bone regeneration" increased the quality of the manuscript significantly and addressed the mentioned issues. Therefore, I suggest the editor to accept this article.

Reviewer 4 Report

Authors have addressed all the major aspects of the manuscript, improving the quality and suitability of the review. 

Just a very minor comments:

Section 3.5. Should be named In vitro and in vivo "studies"